# Fecal and Urinary Adipokines as Disease Biomarkers

**DOI:** 10.3390/biomedicines11041186

**Published:** 2023-04-16

**Authors:** Hauke C. Tews, Tanja Elger, Thomas Grewal, Simon Weidlich, Francesco Vitali, Christa Buechler

**Affiliations:** 1Department of Internal Medicine I, Gastroenterology, Hepatology, Endocrinology, Rheumatology and Infectious Diseases, University Hospital Regensburg, 93053 Regensburg, Germany; hauke.tews@klinik.uni-regensburg.de (H.C.T.); tanja.elger@klinik.uni-regensburg.de (T.E.); 2School of Pharmacy, Faculty of Medicine and Health, University of Sydney, Sydney, NSW 2006, Australia; thomas.grewal@sydney.edu.au; 3Department of Internal Medicine II, School of Medicine, University Hospital Rechts der Isar, Technical University of Munich, 81675 Munich, Germany; simon.weidlich@mri.tum.de; 4Department of Medicine 1, Gastroenterology, Pneumology and Endocrinology, Friedrich-Alexander-Universität Erlangen-Nürnberg, 91054 Erlangen, Germany; francesco.vitali@uk-erlangen.de

**Keywords:** adiponectin, lipocalin-2, leptin, galectin-3, chemerin, interleukin-6, kidney, liver, inflammatory bowel disease, biomarker

## Abstract

The use of biomarkers is of great clinical value for the diagnosis and prognosis of disease and the assessment of treatment efficacy. In this context, adipokines secreted from adipose tissue are of interest, as their elevated circulating levels are associated with a range of metabolic dysfunctions, inflammation, renal and hepatic diseases and cancers. In addition to serum, adipokines can also be detected in the urine and feces, and current experimental evidence on the analysis of fecal and urinary adipokine levels points to their potential as disease biomarkers. This includes increased urinary adiponectin, lipocalin-2, leptin and interleukin-6 (IL-6) levels in renal diseases and an association of elevated urinary chemerin as well as urinary and fecal lipocalin-2 levels with active inflammatory bowel diseases. Urinary IL-6 levels are also upregulated in rheumatoid arthritis and may become an early marker for kidney transplant rejection, while fecal IL-6 levels are increased in decompensated liver cirrhosis and acute gastroenteritis. In addition, galectin-3 levels in urine and stool may emerge as a biomarker for several cancers. With the analysis of urine and feces from patients being cost-efficient and non-invasive, the identification and utilization of adipokine levels as urinary and fecal biomarkers could become a great advantage for disease diagnosis and predicting treatment outcomes. This review article highlights data on the abundance of selected adipokines in urine and feces, underscoring their potential to serve as diagnostic and prognostic biomarkers.

## 1. Introduction

Adipose tissues are composed of adipocytes and stromal vascular cells such as preadipocytes, T cells and monocytes. Adipokines represent a group of cytokines released by adipose tissues irrespective of their cellular origin. Although cells not localized in adipose tissue can contribute to adipokine production, several adipokines are almost exclusively produced by adipocytes, while others mainly originate from stromal vascular cells [1,2,3,4,5,6].

Body fat is classified into white and brown adipose tissue, each with specific physiological functions. The main duty of white adipose tissue is the storage of neutral lipids in the form of cholesteryl esters and triglycerides, whereas the central function of brown fat is the production of heat to maintain body temperature [7,8]. White adipose tissues are organized into subcutaneous and visceral adipose tissues. In addition, pulmonary, gonadal, epicardial, perirenal and bone marrow-associated adipose tissues serve highly specific functions, which have been reviewed in detail [9,10].

The quantity of these secreted bioactive proteins can vary between separate white fat depots, and this can give rise to different biological outcomes [11]. This tissue-specific relationship between adipokine release and biological activity is exemplified by visceral fat (including omental and mesenteric adipose tissues) being associated with adverse metabolic effects such as low-grade inflammation, insulin resistance and hypertension, whereas subcutaneous fat appears rather protective in these settings [12,13]. Interleukin-6 (IL-6) is one of the cytokines differentially produced by subcutaneous and visceral fat, and higher IL-6 release from visceral adipose tissue causes increased portal vein IL-6 levels, which positively correlate with systemic inflammation [14,15].

As adipose tissue-derived cytokines, adipokines have been mostly studied in relation to obesity and metabolic diseases. Most of these bioactive proteins increase in the serum of obese individuals and contribute to low-grade systemic inflammation and dysregulation of glucose and lipid metabolism. Adiponectin and omentin-1 are exceptions, and their circulating levels decline in the obese. These fat-derived cytokines improve metabolic health, and their downregulated expression in fat tissues and circulation is unfavorable in this regard [1,2,3,4,5,6,16,17].

Though not used in clinical routine diagnostics to date, serum levels of distinct adipokines are now well believed to represent excellent biomarkers for metabolic diseases such as insulin resistance or non-alcoholic fatty liver disease (NAFLD) [18,19,20,21]. Furthermore, adipokines can also be found in urine and feces, which are easily accessible and provide an opportunity to establish cost-effective, non-invasive tests. Hence, it has been recognized that determining the levels of specific adipokines in urine and/or feces has the potential for risk stratification in patients with different diseases [22,23,24].

Further support for this concept comes from the increased number of diagnostic tools based on biomarker detection in urine and stool. For instance, urinary albumin and creatinine levels are routine clinical markers for renal dysfunction. The urinary albumin-to-creatinine ratio in early morning urine is used to diagnose albuminuria [25]. The identification of Cushing’s syndrome and Addison’s disease includes the determination of urinary cortisol levels [26]. Alpha1-antitrypsin concentrations in feces are a marker for protein-losing enteropathy [27]. Calprotectin in feces, which is mainly derived from neutrophils, is used to monitor disease activity in inflammatory bowel diseases and, in particular, has high clinical relevance as a surrogate marker in inflammatory bowel disease involving the colon [28,29,30]. The levels of stool lactoferrin, a protein produced by epithelial cells, neutrophils and mononuclear phagocytes, correlate with intestinal inflammation [24]. Likewise, fecal protein panels to discriminate between controls, non-alcoholic steatohepatitis and hepatocellular carcinoma have been recently described [31].

In regard to the utilization of feces for biomarker detection, it has emerged that the gut microbiome can greatly affect the composition of feces, potentially allowing bacterial metabolites to serve as biomarkers for different pathologies. In consideration that disturbed liver and bowel function, and various other disorders such as neurodegenerative diseases, are linked to changes in the microbiome and vice versa, analysis of fecal samples may be attractive for the identification of biomarkers for different diseases [32,33,34,35,36].

A precondition for the identification of urinary and fecal biomarkers is their sufficient abundance in these excretions and the availability of suitable detection assays. Moreover, potential confounding factors such as sex, diet or body mass index (BMI) must be considered. Overall, analysis of the human fecal proteome [37,38,39] identified ~2000 host-derived proteins that overlapped by almost 50% with plasma proteins [38]. For comparison, urine contained ~6000 different proteins, with extracellular and plasma membrane proteins being enriched [40].

Despite intense research, fecal biomarkers currently used in the clinic are still not disease-specific. For example, fecal calprotectin levels are increased in intestinal inflammation, yet this diagnosis is independent of the underlying disease. Moreover, biomarkers to discriminate ulcerative colitis and Crohn’s disease from infectious disorders are still lacking. Yet, based on the rapidly increasing insight within this field, one can envisage that combinations of different biomarkers from serum, urine and stool may improve the diagnostic performance [22,24]. To develop this further, standardized procedures for sample collection and commercial assays will need to be developed that reliably quantify metabolites and various proteins in urine and feces. The necessity to standardize sample collection is apparent when considering urine for biomarker detection, as urine collected in the morning will differ in protein composition compared to urine collected at later times during the day (10–24 h), and their analysis could easily produce discordant results [23]. Other factors contributing to the currently limited utilization of urine and fecal biomarkers include the need for data normalization [23,41,42], adequate test sensitivity and specificity that can discriminate between patients and controls, and defined and established positive and negative predictive values [43].

Despite these limitations, substantial progress has been made in recent years to advance the identification of diagnostic and prognostic biomarkers in urine and stool. It would go beyond the scope of this article to cover all efforts to develop biomarkers based on urine and feces analyses, and we refer the reader to excellent reviews in this field [23,24,44,45,46]. In the following, this review article will provide an overview of urinary and fecal adipokines, which may emerge as biomarkers for different diseases.

Currently available data on urinary and fecal levels of adiponectin, chemerin, galectin-3, IL-6, leptin and lipocalin-2 are summarized. Adiponectin, chemerin and leptin are primarily produced by adipocytes and are the most widely studied adipokines [47,48]. Lipocalin-2, IL-6 and galectin-3 are expressed by adipocytes and immune cells [49,50,51], and a significant number of articles regarding their fecal and urinary levels had been published. A PubMed search for articles containing the term “urinary omentin-1” or “fecal omentin-1” or when omentin was replaced by, e.g., visfatin, vaspin or resistin, did not find any studies. Hence, this review focused on adiponectin, chemerin, galectin-3, IL-6, leptin and lipocalin-2.

## 2. Adipokines in Urine and Feces

### 2.1. Adiponectin

Adiponectin is a very well-studied adipokine with anti-inflammatory, insulin-sensitizing and hepatoprotective activities [52]. Blood concentrations of adiponectin are in the microgram per milliliter range, and ~1000-fold higher than serum levels of most other adipokines. Trimeric, hexameric and high-molecular-weight multimeric adiponectin assemblies circulate in blood, and these complexes differ in their biological activities [52]. Yet, as commercial ELISAs rather measure total adiponectin levels, the concentration of the different adiponectin complexes in various diseases is less well described. Circulating adiponectin levels show a gender disparity and are higher in females than males. Adiponectin is almost exclusively produced in adipocytes, and in the obese, reduced expression in fat cells correlates with a decline in adiponectin serum levels [21,53]. Low adiponectin levels in obesity contribute to insulin resistance and NAFLD [52,54,55,56,57] and are a risk factor for malignant diseases, making adiponectin receptor agonists potential drug candidates for cancer treatment [58].

On the other hand, serum adiponectin levels are elevated in severe liver dysfunction, in cholestasis and in mouse models for liver fibrosis after bile duct ligation. Altogether, these results suggested biliary excretion of adiponectin, which was accordingly detected in human bile [2]. Circulating adiponectin amounts are also increased in chronic kidney diseases, possibly due to impaired renal excretion, pointing at serum and urinary adiponectin levels as potential biomarkers for renal function [59,60].

Hence, serum adiponectin levels are a promising biomarker in metabolic syndrome and kidney and liver diseases and a possible therapeutic target in cancer. In the following, we will list currently available data on adiponectin levels in the urine and feces in examples of metabolic, kidney and cancer diseases.

#### 2.1.1. Urinary Adiponectin in Diabetic Patients

Adiponectin levels in urine are ~1000-fold lower compared to those in the circulation (see above) [59,61], with high- (octadecamer), middle- (hexamer) and low-molecular-weight (trimer) adiponectin being detectable in urine [62].

Diabetic kidney disease is a common complication of type 1 and type 2 diabetes [63]. Total and high-molecular-weight adiponectin correlated with measures of renal dysfunction in diabetic patients [64]. In the urine of patients with kidney diseases of different etiologies, low-molecular-weight adiponectin levels were approximately 2-fold higher than the levels of the high-molecular-weight form, and both isoforms were strongly increased in proteinuric patients [65]. Urinary adiponectin concentrations of diabetic patients were higher compared to obese and normal weight controls, with no differences between the latter two groups. Adiponectin concentrations in urine negatively correlated with renal function, and low-molecular-weight adiponectin became more abundant when the estimated glomerular filtration rate (eGFR) declined. Most interestingly, urinary adiponectin levels were already elevated at a timepoint when urinary albumin levels were still unchanged, which may point to adiponectin urine levels as an early marker for renal dysfunction [62]. Middle- and high-molecular-weight adiponectin amounts in urine were related to blood glucose concentrations [62]. Higher urinary and serum adiponectin levels were also observed in type 2 diabetic patients with macroalbuminuria. Serum and urinary adiponectin levels were elevated ~1.5-fold and ~10-fold, respectively, suggesting that higher adiponectin systemic levels are probably not the sole reason for excess amounts of adiponectin in urine [66] (Figure 1).

Prospective studies showed that urinary high-molecular-weight adiponectin levels predicted a decline in renal function in patients with type 2 diabetes [67,68]. Yet, serum adiponectin concentrations were mostly low in type 2 diabetics but increased in type 1 diabetics [69,70], indicating the need for further research to evaluate the suitability of urinary adiponectin levels as a predictive biomarker for renal impairment in type 1 diabetes.

Urinary adiponectin levels were also found ~2-fold increased in type 2 diabetics vs. controls. Carotid artery intima-media-thickness as a measure of atherosclerotic disease positively correlated with urinary adiponectin concentrations in type 2 diabetic patients, indicating urinary adiponectin as a biomarker for vascular damage in type 2 diabetes (Figure 1) [71]. A clearer picture still has to emerge, as another cohort reported urinary adiponectin levels not discriminating between women with and without metabolic syndrome [72].

In summary, and substantiating a relationship also documented for serum adiponectin, urinary adiponectin levels are elevated in diabetic patients with renal diseases (Figure 1). In this context, the determination of adiponectin levels in urine could close a diagnostic gap, as the recognition of kidney dysfunction at an early stage is of decisive importance for risk stratification of cardiovascular diseases and therapy optimization.

#### 2.1.2. Urinary Adiponectin in Systemic Lupus Erythematosus (SLE)

Renal impairment is also common in SLE, a chronic autoimmune disease. Lupus nephritis is present in 15–30% of these patients at the time of diagnosis, and the incidence increases to 30–50% during disease progression [46]. In lupus nephritis, adiponectin was found on the surface of podocytes and in the tubules of the kidneys. Patients with renal impairment had ~7-fold higher urinary adiponectin levels compared to controls [59,61] (Figure 1). Urinary adiponectin levels positively correlated with plasma adiponectin concentrations, which were also increased in lupus nephritis [59]. Furthermore, urinary adiponectin amounts declined in patients with complete remission [73].

Renal diseases increase the risk for thromboembolic events, which are significantly more frequent in SLE [74]. Accordingly, a better understanding of adiponectin function in these settings and the determination of urinary adiponectin levels could be helpful in assessing risk factors for poor outcomes in SLE patients.

#### 2.1.3. Urinary Adiponectin in Patients with Focal Segmental Glomerulosclerosis

Focal segmental glomerulosclerosis (FSGS) is a rare disease characterized by the scarring of the kidney glomeruli [75]. Serum as well as urine adiponectin levels correlated with proteinuria in FSGS patients (Figure 1). However, response to therapy in FSGS patients was not associated with a change in adiponectin levels in serum or urine [76].

#### 2.1.4. Fecal Adiponectin in Colorectal Cancer

A large number of preclinical studies associated low adiponectin levels with a higher risk for malignant diseases [58]. Although this hypothesis was supported by a meta-analysis of colorectal cancer, this could not be confirmed by another meta-analysis of case–control studies [77,78]. Nevertheless, experimental work supported the protective activities of adiponectin in colorectal cancers [79,80]. Accordingly, most studies showed low serum adiponectin levels in patients with colorectal cancers compared to healthy controls [81]. However, patients with colorectal liver metastases displayed higher serum adiponectin levels than the respective controls, suggesting that adiponectin serum levels increased during cancer progression [82].

Despite these promising findings to develop serum adiponectin levels as a diagnostic tool for certain cancers, major challenges of adiponectin-related association studies remain, as serum adiponectin levels are related to gender, BMI and metabolic diseases such as hypertension and liver steatosis [18,21,56,83,84]. Thus, it remains challenging to design studies that can account for all these confounding factors, and it remains to be determined if the association of blood adiponectin levels and colorectal carcinoma is influenced by any of these traits.

Alternatively, a screen for elevated proteins in the stool of patients with colorectal cancers identified matrix metalloprotease 9, haptoglobin, myeloperoxidase and fibrinogen, as well as adiponectin [85] (Figure 1). The upregulated amounts of these proteins discriminated colorectal cancer from adenoma and healthy controls. Serum adiponectin levels were not determined in this study, and correlations with fecal adiponectin levels could not be calculated [85]. These promising observations demand confirmation in other cohorts, and further evidence would also be needed to determine if fecal adiponectin levels have the capacity to predict drug therapy response in colorectal cancer [86].

### 2.2. Lipocalin-2

Lipocalin-2 (also known as neutrophil gelatinase-associated lipocalin) is an anti-inflammatory protein and reduces lipopolysaccharide-induced cytokine release [51]. Unraveling the multiple biological activities of lipocalin-2 has been challenging, as three lipocalin-2 variants with differential expression patterns and serum levels have been reported to date [87]. In addition, mono- and dimeric forms of lipocalin-2 with potentially different functions exist [88]. Variants include the polyaminated lipocalin-2, non-polyaminated C87A, and R81E variants, all with specific ligand-binding activities. All three variants were higher in the serum and urine of healthy males compared to healthy females [87], and accordingly, higher levels of total lipocalin-2 in males than in females have been reported [89]. Most relevant for diagnostic purposes, all three variants display distinct expression patterns in cardiovascular and metabolic disorders, with variant-specific roles in cardiovascular, renal and metabolic malfunctions. However, currently available patient-derived data have predominantly documented only total lipocalin-2 levels [87].

Lipocalin-2 levels are induced in response to inflammatory factors in different cell types such as lymphocytes, adipocytes and hepatocytes. In addition, neutrophils can also store and, upon stimulation, release lipocalin-2 from secretory vesicles [90]. Lipocalin-2 is upregulated in adipose tissues and the liver in obesity [51,91], and as lipocalin-2 decreases food intake and promotes fat cell browning, this indicates higher lipocalin-2 levels to counteract obesity and metabolic diseases [92,93]. Furthermore, lipocalin-2 scavenges iron, providing this adipokine with bacteriostatic properties, as iron depletion interferes with bacterial growth [94].

An association of circulating lipocalin-2 levels with BMI has been observed in some, but not all studies [93,95,96,97]. In addition, lipocalin-2 serum levels were higher in patients suffering from hypertension [98,99] and were reported to possibly decline in type 2 diabetes patients [99,100], indicating a complex interplay between metabolic diseases and serum lipocalin-2 levels.

Besides metabolic disorders, lipocalin-2 also has a role in a variety of cancers, and tumor-promoting functions such as increased cell proliferation and reduced apoptosis have been described [101]. In cholangiocarcinoma, lipocalin-2 promoted tumor growth, and its expression was negatively associated with patient survival [102]. Biliary lipocalin-2 levels of cholangiocarcinoma patients were higher in comparison to those of patients with gallstones, suggesting that lipocalin-2 was excreted by the liver [102]. This observation might indicate a diagnostic and predictive value of fecal lipocalin-2 levels for cholangiocarcinoma patients (Figure 2). Intrahepatic cholangiocarcinoma represents a very heterogeneous cancer type with a variety of drug-addressable genetic mutations [103]. Consequently, fecal lipocalin-2 levels may become a valuable parameter not only for tumor progression, but also for treatment efficacy.

In urine, lipocalin-2 levels were reported to be low as most of the circulating lipocalin-2 appears to be captured by proximal tubular epithelial cells. Thus, urinary lipocalin-2 may originate from serum, or from injured tubular cells, which themselves produce lipocalin-2. The clearance of lipocalin-2 also differs between its monomeric and dimeric isoform, and physiologically, the monomer is more rapidly excreted. Monomeric lipocalin-2 was released by human kidney 2 (HK-2) cells, which mimic proximal tubular kidney cells, while the dimer was secreted by neutrophils. Accordingly, in urine, the dimer was the predominant form in patients with urinary tract infections, but higher monomer concentrations were found in patients with acute kidney injury [88,104]. Serum lipocalin-2, but not urinary lipocalin-2, correlated with BMI, blood pressure, heart rate, triglycerides and high-sensitivity C-reactive protein. Urinary lipocalin-2 variants were associated with measures of kidney function [87]. Of note, the limited stability of lipocalin-2 in urine or other fluids over extended periods of storage may need further data analysis to substantiate these findings [105].

#### 2.2.1. Urinary Lipocalin-2 in SLE

The autoimmune disease SLE are triggered by autoantibodies against double-stranded DNA affecting renal structures. Exposure of kidney mesangial cells to these pathogenic autoantibodies strongly upregulated lipocalin-2 expression [106], which raised the possibility that urinary lipocalin-2 levels could serve as a biomarker to discriminate between SLE patients with or without lupus nephritis. Indeed, SLE patients with lupus nephritis had higher urinary lipocalin-2 levels compared to those SLE patients without kidney disease [107]. Along these lines, a recent meta-analysis found urinary lipocalin-2 levels suitable for the detection of lupus nephritis (Figure 2). Furthermore, urinary levels of lipocalin-2 were related to disease activity, indicating the potential to predict renal flare. Notably, serum lipocalin-2 levels appeared not suitable as a diagnostic biomarker for lupus nephritis [108]. In childhood-onset SLE, urinary lipocalin-2 levels were a highly sensitive and specific disease marker, clearly discriminating against juvenile idiopathic arthritis [109]. In summary, as early diagnosis of lupus nephritis lowers the risk for progression to end-stage renal disease and death, the determination of urinary lipocalin-2 levels could become a suitable biomarker for kidney function.

#### 2.2.2. Urinary Lipocalin-2 in Diabetic Chronic Kidney Disease

Diabetes is a risk factor for impaired kidney function. In diabetic chronic kidney disease, urinary lipocalin-2 levels were ~5-fold higher compared to controls (Figure 2). Urinary lipocalin-2 concentrations also positively correlated with urinary protein content and microalbumin, and inversely with eGFR [110].

Urinary lipocalin-2 levels did not differ between metabolically healthy controls, newly diagnosed type 2 diabetics and long-standing diabetics without clinical evidence of diabetic kidney disease. An increase in urinary lipocalin-2 amounts was noticed in diabetic patients with clinical signs of renal diseases. In the cohort of diabetic patients, urinary lipocalin-2 levels positively correlated with serum creatinine, cystatin C, hemoglobin A1c (HbA1c) and albuminuria [111].

In addition, lipocalin-2 levels measured in first morning urine were associated with cardiovascular complications, independent of established risk factors and diabetes duration. One year of therapy led to improved HbA1c levels and eGFR scores and correlated with a decline in urinary lipocalin-2 levels [112].

#### 2.2.3. Urinary Lipocalin-2 in Patients with Macroproteinuria, Glomerulonephritis, Kidney Transplantation and Sepsis

Persistent proteinuria is an indicator of serious renal injury and is caused by various diseases. Urinary lipocalin-2 levels of 33 patients with persistent macroproteinuria (proteinuria >1 g/day for at least 6 months) were more than 5-fold higher compared to those from 20 healthy subjects. Urinary lipocalin-2 amounts positively correlated with urinary protein loss and were inversely correlated to residual renal function. Hence, detectable amounts of urinary lipocalin-2 appear to originate from serum and reflect impaired glomerular function [113].

Urinary lipocalin-2 is also associated with renal dysfunction and predicted progression of renal injury in primary glomerulonephritis, a rare disease that can cause chronic kidney disease and frequently affects young people [114,115].

Kidney transplantation is the preferred treatment for patients with end-stage renal diseases. Urinary lipocalin-2 levels predicted the change in eGFR during the one-year follow-up in patients who received a kidney transplant (Figure 2) [116]. Importantly, urinary lipocalin-2 levels of kidney donors were not associated with organ rejection or graft failure [117].

Critically ill patients with acute kidney injury also displayed high urinary lipocalin-2 levels, and this was related to disease progression and mortality [118]. However, with urinary lipocalin-2 levels being elevated in patients with sepsis, urinary infections or cardiopulmonary bypass surgeries, the prognostic utility of urinary lipocalin-2 concentrations for acute kidney injury appears limited, which is especially relevant for patients in intensive care [119].

#### 2.2.4. Urinary Lipocalin-2 in Inflammatory Bowel Disease

Out of the inflammatory bowel diseases, Crohn’s disease can affect any segment of the gastrointestinal tract. In contrast, ulcerative colitis represents a continuous inflammation from the distal to the proximal colon without affecting the upper gastrointestinal tract. Renal complications are one of the extraintestinal manifestations of these diseases [120]. Patients with active disease exhibited higher microalbuminuria than patients with inactive disease, which still presented increased urinary albumin compared to healthy controls [121]. Consistent with these results, Crohn’s disease patients showed almost 30-fold higher urinary lipocalin-2 levels compared to controls (Figure 2). A single high dose of infliximab, which is an anti-tumor necrosis factor antibody widely used to treat these patients, significantly lowered urinary lipocalin-2 levels by more than 60% [122,123].

Despite these promising results, indicating the potential for urinary lipocalin-2 levels as a biomarker for Crohn’s disease, further research is still needed, as another study showed that serum, but not urinary, lipocalin-2 levels were elevated in inflammatory bowel disease and correlated with disease activity [124].

#### 2.2.5. Urinary Lipocalin-2 in Pancreatic Diseases

Pancreatic diseases such as acute and chronic pancreatitis and pancreatic ductal adenocarcinoma have in several cases been linked to increased lipocalin-2 expression [125]. Comparing urinary lipocalin-2 amounts in small cohorts of focal chronic pancreatitis and pancreatic adenocarcinoma revealed ~2-fold higher levels in the latter [126]. In addition, higher urinary lipocalin-2 levels were related to disease severity and predicted survival of patients with acute pancreatitis [127]. On the other hand, a second cohort of patients with acute pancreatitis did not reveal differences in urinary lipocalin-2 levels compared with the reference group [128].

#### 2.2.6. Urinary Lipocalin-2 in Liver Cirrhosis

Acute kidney injury is a common complication of decompensated liver cirrhosis and bears a high mortality risk. Most interestingly, urinary lipocalin-2 levels were low in pre-renal acute kidney injuries, higher in hepatorenal syndrome and highest in acute tubular necrosis, which are the main causes of acute liver injury in liver cirrhosis [129,130] (Figure 2). This could point to urinary lipocalin-2 levels differentiating between the causes of acute kidney injury in liver cirrhosis. In fact, in the patient cohort described above, urinary lipocalin-2 levels better predicted 90-day transplant-free survival than the Model for End-Stage Liver Disease score [130].

#### 2.2.7. Fecal Lipocalin-2 in Necrotizing Enterocolitis

Necrotizing enterocolitis is a life-threatening intestinal disease of premature infants. Further complicating treatment options, it remains difficult to distinguish necrotizing enterocolitis from other intestinal diseases [131]. As early diagnosis of this disease allows targeted intervention such as antibiosis, a non-invasive and rapid diagnosis still has to be developed. In the search for a non-invasive biomarker, fecal calprotectin levels were found to be elevated in infants with necrotizing enterocolitis, with increased levels already detectable in the stool 7 to 10 days prior to current diagnosis, which relies on radiologic and clinical characteristics. In addition, fecal lipocalin-2 levels were also suggested as a potential non-invasive marker of intestinal inflammatory diseases [132]. Indeed, lipocalin-2 concentrations were induced in fecal samples of preterm infants before the diagnosis of necrotizing enterocolitis [131,133] (Figure 2).

#### 2.2.8. Fecal Lipocalin-2 in Inflammatory Bowel Disease

As outlined above, ulcerative colitis and Crohn’s disease represent the two main entities of inflammatory bowel disease [123]. Calprotectin released by granulocytes was higher in the stool of patients with intestinal inflammation but was not a specific marker for inflammatory bowel disease [29,44]. Nevertheless, fecal calprotectin levels were a useful biomarker to monitor intestinal inflammation and to predict disease relapse [22,134].

Measurement of additional biomarkers may improve the diagnosis of inflammatory bowel diseases, and determining fecal lipocalin-2 levels may represent an attractive option [135]. Fecal lipocalin-2 levels were elevated ~4–5-fold in active ulcerative colitis and active Crohn’s disease compared to patients with inactive disease, the latter still being ~4–5-fold higher compared to stool from healthy controls (Figure 2). Fecal lipocalin-2 levels positively correlated with fecal calprotectin amounts and, importantly, with several other established diagnostic measurements, such as ulcerative colitis Mayo endoscopic score, ulcerative colitis Mayo partial score, Crohn’s Disease Endoscopic Index of Severity and Harvey–Bradshaw index [135]. Similar trends were provided by others, documenting higher fecal lipocalin-2 levels in active compared to quiescent inflammatory bowel diseases [136]. In these studies, serum levels of lipocalin-2 did not differ between quiescent and active stages of inflammatory bowel diseases, suggesting that elevated fecal lipocalin-2 levels did not originate from serum [136]. Likewise, in another cohort of inflammatory bowel diseases, fecal lipocalin-2 levels positively correlated with disease severity and endoscopic activity scores [137]. In asymptomatic ulcerative colitis patients, fecal lipocalin-2 levels below a certain threshold (12 mg/kg) ruled out microscopic and macroscopic lesions with high probability (75–93%). Levels above this value were, however, unreliable as predictive biomarkers [138]. Fecal lipocalin-2 levels were approximately 5 mg/kg in active inflammatory bowel disease [135]. While these findings underscore the diagnostic value of fecal lipocalin-2 levels, it is important to note that the methodology that would allow standardization of quantitative analysis of fecal lipocalin-2 levels remains to be established. Yet, while absolute values for fecal lipocalin-2 levels from different laboratories may not be comparable, the abovementioned studies with comparisons relative to appropriate controls point to great biomarker potential.

Finally, showing promise to distinguish between different inflammatory diseases, fecal lipocalin-2 concentrations were reported to be higher in active ulcerative colitis compared to active Crohn’s disease [139]. On the other hand, lipocalin-2 levels were comparable in the stool of patients with infectious enterocolitis and inflammatory bowel diseases [135]. Hence, fecal lipocalin levels may discriminate between some, but not all, different inflammatory bowel diseases.

In summary, several studies revealed promise to develop fecal lipocalin-2 levels as a biomarker for inflammatory bowel diseases. With the development of standardized protocols in future studies, one can envisage that the additional analysis of fecal lipocalin-2 levels could provide a diagnostic and prognostic advantage in comparison to the sole analysis of fecal calprotectin concentrations in inflammatory bowel diseases.

#### 2.2.9. Fecal Lipocalin-2 in Irritable Bowel Disease

Irritable bowel syndrome is a chronic gastrointestinal disease with abdominal pain and altered frequency of stool without an organic reason [140]. Patients with irritable bowel syndrome and healthy controls had comparable fecal lipocalin-2 levels [135]. In contrast, a second study documented higher fecal lipocalin-2 levels in irritable bowel disease compared to healthy controls [136] (Figure 2). Ethosuximide, an anti-epileptic drug, given for 12 weeks improved abdominal pain, lowered systemic inflammatory markers and correlated with reduced fecal lipocalin-2 levels [141].

#### 2.2.10. Fecal lipocalin-2 in Parkinson’s Disease

Lipocalin-2 concentrations in plasma, but not feces, were found to be increased in Parkinson’s disease patients [142,143]. The latter study showed that fecal lipocalin-2 levels were associated with microbial beta diversity, which describes differential microbiome composition between two communities. Thus, fecal lipocalin-2 levels may emerge as a marker of microbiome dissimilarities (Figure 2) [142], which are associated with neurological, tumor and liver diseases [32,144,145].

### 2.3. Leptin

Leptin was initially identified as a satiety hormone with low leptin levels or leptin deficiency causing obesity. Yet, serum leptin levels are increased in the obese, suggesting the development of leptin resistance, which seems to be mainly caused by hyperinsulinemia [146]. In addition, leptin has a role in fertility, inflammation and tissue fibrosis [147,148,149] and was identified as a risk factor for some cancers independent of obesity [150]. The biological activities of leptin are mediated through its binding to the leptin receptor (ObR), which is a class I cytokine receptor. To date, six leptin receptor isoforms have been identified and include a long isoform, four short isoforms and a soluble isoform. The latter can still bind leptin and therefore effectively affects leptin bioavailability [149].

Women commonly display higher serum leptin levels than men, and serum leptin levels increase with BMI in both genders [151]. Renal degradation and urinary excretion are the main elimination routes for leptin. Based on the comparison of arterial and renal vein leptin concentrations of normal postabsorptive volunteers, it was calculated that ~80% of plasma leptin is taken up by the kidney. In earlier studies, leptin in urine could only be detected in a few cases and was generally below the detection limit of the assays used at the time. Based on these findings, it was concluded that the vast majority of leptin molecules are degraded in the kidney [152]. More recently, and utilizing a more sensitive assay with a detection limit in the picogram per milliliter range, compared to the nanogram per milliliter range of leptin commonly observed in plasma, leptin in human urine was detected [153]. Similar to the sex-specific differences in leptin levels in plasma (see above), leptin concentrations were also elevated in women’s urine [153,154,155] and correlated with age and body weight [156].

#### 2.3.1. Leptin in Proteinuria and Nephrotic Syndrome

Leptin levels in the urine of proteinuric children were higher in comparison to healthy controls (Figure 3). Urinary leptin concentrations did not differ between non-proteinuric patients and those patients without nephrotic syndrome [157]. In addition, urinary leptin levels were markedly higher in a cohort of 30 children with nephrotic syndrome compared to 25 healthy controls, and greatly improved during prednisone therapy [158]. While these observations may implicate biomarker potential for urinary leptin levels in some disease phenotypes related to kidney dysfunction, it should be noted that serum leptin levels of control, proteinuria and nephrotic syndrome within these cohorts were comparable, and the underlying cause for elevated urinary leptin levels in these diseases has yet to be identified [157,158].

One possible explanation for the above-mentioned observation could be related to increased levels of soluble lectin receptors, which correlated with elevated leptin levels in urinary samples of another cohort of children with proteinuria aged between 6 and 12 years [159]. In this study group, serum leptin levels were lower in proteinuric children in comparison to controls, indicating an induction of renal clearance [159].

#### 2.3.2. Leptin in Urinary Tract Infection

A potential role for urinary leptin as a biomarker for treatment efficacy may also exist in urinary tract infections of children, as leptin levels in the urine of children with urinary tract infections declined during therapy (Figure 3). Similar to several studies described above (see Section 2.3.1), this decline was not related to a change in mean serum leptin levels [156]. These findings may indicate that drug-mediated remission of the infection in the urinary tract leads to the normalization of urinary leptin levels. However, given the strongly increased sensitivity of currently available detection assays for leptin, it has yet to be determined if significant changes observed in the picogram per milliliter range for urinary leptin levels can be related to measurable and statistically significant differences in leptin concentrations in plasma, which are commonly ~1000-fold higher compared to urine and can range from 0.4 to 50 ng/mL in men and women of control groups, respectively [151].

#### 2.3.3. Leptin in Obstructive Sleep Apnea Syndrome

Obesity is a main risk factor for obstructive sleep apnea syndrome. Upper airway obstruction, fragmented sleep and chronic nocturnal intermittent hypoxia are the manifestations of this disease [160]. Some of these symptoms might be related to leptin, as a recent meta-analysis documented elevated serum leptin levels in patients with obstructive sleep apnea syndrome compared to controls [160]. Notably, serum and urine leptin levels positively correlated in patients and controls when collected overnight or during the day [153]. These findings markedly differ from observations from kidney dysfunctions or urinary tract infections described above (Section 2.3.1 and Section 2.3.2) and suggest that urinary leptin levels appear as appropriate as serum leptin levels to characterize obstructive sleep apnea syndrome and possibly other diseases (Figure 3) [155].

#### 2.3.4. Leptin in Feces

The amount of leptin released by subcutaneous fat is generally 2–3-fold higher than leptin secreted by visceral adipose tissues [161]. Accordingly, leptin concentrations in systemic blood are higher than those in the portal and hepatic veins, which both display comparable levels [2], strongly indicating that the excretion of serum-derived leptin from the liver is marginal.

In line with this observation, leptin has rarely been analyzed or even detected in stool. With the exception of one study that was able to detect leptin in the feces of 3 out of 24 healthy controls [162], to our knowledge, leptin levels have not been analyzed systematically in fecal samples of patients (Figure 3).

### 2.4. Galectin-3

Galectin-3 is an inflammatory and fibrotic molecule expressed by all immune cells as well as epithelial and endothelial cells [163]. Adipocytes also express galectin-3, and circulating levels of galectin-3 are increased in obesity [164,165]. Extracellular galectin-3 regulates the interaction of cells and modulates the association of epithelial cells with extracellular matrix proteins [163].

In the liver, galectin-3 expression was strongly induced in cirrhosis [166,167,168], and higher serum galectin-3 levels were reported in patients with chronic liver diseases of different etiologies when compared to liver-healthy controls [167,169]. Circulating galectin-3 levels were higher in type 2 diabetes, heart failure, macroalbuminuria, cardiovascular diseases and certain types of cancer such as colorectal and gastric cancer [163,170].

Given its involvement in multiple pathological settings, galectin-3 has become a therapeutic target, and genetic or pharmacological inhibition of galectin-3 improved renal dysfunction in different pathological conditions [171] and was also protective in experimental models of liver fibrosis [168,172,173].

#### 2.4.1. Urinary Galectin-3 and Kidney Fibrosis

Renal fibrosis is the end stage of progressive kidney diseases and is clinically verified after biopsy, an invasive procedure often associated with adverse events. Alternatively, the determination of plasma levels of fibrosis markers could be considered. However, they often lack specificity to provide information on organ-specific disease states [171,174]. Hence, the development of urine biomarkers to determine kidney fibrosis may be more appropriate for routine tests in the clinic or research-related and laboratory-based studies.

A prospective study showed that patients with higher amounts of urinary galectin-3 (≥510.8 pg/mL) compared to those with lower galectin-3 levels (≤354.6 pg/mL) had a 4.6-fold higher risk of kidney disease progression (Figure 4). Furthermore, renal galectin-3 gene expression was positively associated with biomarkers of renal stress, renal fibrosis and kidney dysfunction [174]. Given its involvement in inflammatory and fibrotic processes, the determination of urinary galectin-3 levels may become a non-invasive biomarker for advanced kidney diseases.

#### 2.4.2. Urinary Galectin-3 and Heart Failure

In patients with heart failure, mean plasma galectin-3 levels were 1.7-fold higher compared to healthy controls, while urinary galectin-3 levels in these two groups were comparable. As renal excretion of galectin-3 of patients was slightly lower compared to controls, it was proposed that impaired renal excretion of galectin-3 may contribute to higher plasma galectin-3 levels observed in patients. This hypothesis is supported by the very high plasma galectin-3 levels found in dialysis patients who were anuric [175]. It was also reported that high urinary galectin-3 levels in patients with heart failure were a prognostic factor for renal dysfunction and death [176]. Hence, urinary galectin-3 levels may serve as a prognostic biomarker for fatal outcome in patients with cardiovascular diseases (Figure 4).

#### 2.4.3. Urinary Galectin-3 and Cancer

Urinary galectin-3 concentrations were found to increase in a variety of tumors including cervical, esophageal and breast cancer. Higher galectin-3 concentrations were also detected in urine samples from a mixed cohort of patients with Hodgkin’s lymphoma, liver cancer or multiple myeloma. Moreover, urinary galectin-3 levels increased with tumor stage, showing that galectin-3 secretion was enhanced during tumor progression (Figure 4). Importantly, a decline in urinary galectin-3 levels was noticeable 5 days after initiation of therapy, indicating that urinary galectin-3 levels might be suitable to serve as a read-out for treatment efficacy [177].

#### 2.4.4. Fecal Galectin-3 and Cancer

Galectin-3 is also highly abundant in stool, and fecal galectin-3 levels were increased in colorectal carcinoma patients with advanced tumor node metastasis (TNM) score, higher nuclear grade, poor tumor tissue differentiation and metastatic disease (Figure 4). Positive correlations between fecal galectin-3 concentrations and the cancer biomarkers alpha-fetoprotein and carcinoembryonic antigen were also observed [178]. Thus, the determination of not only urinary but also fecal galectin-3 levels might provide useful clinical information in advanced colorectal cancers.

#### 2.4.5. Fecal Galectin-3 and Inflammatory Bowel Disease

The addition of dextran sodium sulfate in drinking water is a common experimental approach to trigger inflammatory bowel disease in mice. When comparing wild-type and galectin-3-deficent mice in this model for inflammatory bowel disease, galectin-3 contributed to the resolution of inflammation, and acute dextran sodium sulfate-induced colitis was ameliorated by galectin-3 treatment [179]. In acute as well as chronic colitis models, peritoneal administration of recombinant galectin-3 caused a significant decline in colonic IL-6 levels [180].

Notably, galectin-3 levels in serum and stool of patients with ulcerative colitis negatively correlated with endoscopic and histological parameters of colitis. Elevated fecal galectin-3 levels (>553.44 pg/mL) were an indicator of the attenuation of ulcerative colitis [179] (Figure 4).

Fecal galectin-3 levels of patients with ulcerative colitis and metabolic syndrome were ~3-fold higher compared to those of ulcerative colitis patients not suffering from metabolic dysfunction. The latter group had more severe bowel disease as assessed clinically and histologically [181]. Fecal galectin-3 levels have the potential to become a non-invasive biomarker for the diagnosis and prognosis of inflammatory bowel diseases.

### 2.5. Chemerin

Chemerin is a chemoattractant protein abundantly expressed in adipocytes and hepatocytes. In addition, chemerin expression, albeit at much lower levels, has been identified in various other cell types such as epithelial and endothelial cells [48,182,183]. Increased circulating chemerin levels in obesity correlate with systemic inflammation [184]. The regulation of secreted chemerin is complex, as cells release biologically inactive chemerin, which has to be activated by C-terminal proteolysis [185]. In fact, most of the chemerin circulating in the blood is inactive and requires conversion to active forms by different proteases [48,186]. Once activated, chemerin is a chemoattractant for cells expressing chemokine-like receptor 1 (CMKLR1) such as monocytes, natural killer cells or dendritic cells. Through the chemerin-mediated recruitment of these immune cells at sites of infections, chemerin actively participates in the resolution of inflammation. Additionally, chemerin has been assigned roles in insulin sensitivity, adiposity and blood pressure control, which has been reviewed in detail [48,187,188]. Furthermore, more recent findings indicate chemerin exerts both tumor-promoting and -suppressive functions, depending on the tumor analyzed. In line with these observations, chemerin is either increased or reduced in tumor tissues in comparison to tumor-adjacent tissues [189,190,191].

There is some evidence that liver-released chemerin adds to its serum levels, and serum chemerin concentrations were indeed low in patients with liver cirrhosis [2,192,193]. Patients with kidney dysfunction displayed higher serum chemerin levels [194], which may be partly due to impaired renal excretion of this chemokine.

#### Urinary and Fecal Chemerin in Inflammatory Bowel Disease

Chemerin protein can be detected in rodent urine using a commercial enzyme-linked immunosorbent assay (ELISA) [195], and another study examined urinary chemerin levels in more detail in patients with inflammatory bowel disease. Urinary chemerin protein levels did not correlate with serum chemerin amounts, BMI or age and did not differ between sexes. Chemerin concentrations in the urine of patients with inflammatory bowel disease were substantially smaller (20–1470 pg/mL; mean 34 pg/mL) compared to serum chemerin levels (82–391 ng/mL; mean 190 ng/mL) [196].

Urinary chemerin levels of healthy controls and patients with inflammatory bowel disease and lower fecal calprotectin levels (<500 µg/g) were comparable. However, showing some biomarker potential in selected patients, inflammatory bowel disease characterized by high fecal calprotectin levels (>500 µg/g) also showed elevated urinary chemerin concentrations in comparison to patients with calprotectin levels below this threshold or healthy controls [196].

Immunoblot analysis of human urine detected chemerin protein with a molecular weight of ~15 kDa, which corresponded to full-length chemerin and showed that intact—or mostly intact—protein was excreted by the kidney. It remains to be determined if kidney cells produce and secrete chemerin and thereby contribute to chemerin urinary levels [197]. In further support of its diagnostic potential, chemerin levels in urine negatively correlated with serum creatinine [196], which is a reasonable estimate of renal function [198].

Chemerin protein could be detected in the stool of 20% of the patients by immunoblot, and ELISAs confirmed low chemerin concentrations in the majority of feces samples, making it unsuitable to develop further as a diagnostic tool [196]. Yet, based on the findings described above, urinary chemerin levels may emerge as a further biomarker for certain patient groups suffering from bowel inflammation.

### 2.6. Interleukin-6 (IL-6)

IL-6 is a multifunctional cytokine produced in response to infections or injuries and contributes to a range of immune, hematopoietic and acute-phase responses as part of the host defense. IL-6 is produced by a variety of cells including T cells, monocytes, fibroblasts and adipocytes and induces inflammatory biomarkers such as C-reactive protein. IL-6 serum levels are upregulated rapidly after infections, but also in several renal and liver diseases. In addition, IL-6 levels are elevated in obesity with a role in the low-grade systemic inflammation commonly observed in obese people [14,15,199]. Diurnal variations of circulating IL-6 levels have been described, and a meta-analysis stated that the lowest IL-6 levels are found in the morning [200]. In 10 males, serum IL-6 concentrations were higher throughout the night, and IL-6 levels were lower throughout the day. Interestingly, IL-6 concentrations in urine were opposite to these findings, with the highest urinary IL-6 levels during the day and the lowest overnight [201].

Numerous experimental and clinical studies proved the pathological functions of IL-6 in autoimmunity, inflammation and cancer. IL-6 acts through binding to the IL-6 receptor, and tocilizumab is an anti-IL-6 receptor antibody that is well documented to be effective in patients with autoimmune and inflammatory diseases [202]. Moreover, IL-6 can function as a growth factor, and it promotes cancer progression by regulating multiple signaling pathways [203].

#### 2.6.1. Urinary IL-6 in Rheumatoid Arthritis

Rheumatoid arthritis is an autoimmune disease primarily affecting joints [204]. Chronic inflammation causes swelling of the joints and contributes to bone erosion, cartilage destruction and deformity [205].

Patients with rheumatoid arthritis had ~4-fold higher levels of urinary IL-6 in comparison to healthy controls. Urinary IL-6 levels did not differ between patients with low and medium Disease Activity Score 28 scores (<3.2 and <5.1, respectively), but IL-6 concentrations were strongly induced in the urine of patients with a score ≥ 5.1 (Figure 5). Further highlighting its diagnostic potential, urinary IL-6 levels at the time of diagnosis were an independent risk factor for 3-year radiographic progression of the disease. The underlying cause remains to be fully understood, but higher IL-6 levels in the urine of patients with rheumatoid arthritis have been considered to originate from subclinical renal injury and/or may be related to serum IL-6 levels, which are increased in these patients. In fact, in rheumatoid arthritis, urinary IL-6 levels correlated positively with its serum levels as well as serum C-reactive protein, Disease Activity Score 28 and white blood cell count. Relevant for the standardization of protocols, the analysis of first or second morning urine samples seems to be appropriate [206]. In summary, urinary IL-6 levels appear to be a surrogate marker for serum IL-6 concentrations, and sequential monitoring of urinary IL-6 levels may be used to evaluate drug response aiming to lower autoimmune reactions in rheumatoid arthritis.

#### 2.6.2. Urinary IL-6 and Urinary Tract Infection

Children with urinary tract infections had grossly elevated urinary IL-6 levels compared to children without bacteriuria. Urinary IL-6 levels were related to the degree of proteinuria, hematuria and urinary leukocyte counts. In contrast to rheumatoid arthritis, where urinary and serum IL-6 levels correlated (see Section 2.6.1) [206], urinary and serum IL-6 concentrations did not correlate in children with urinary tract infections. Hence, children with urinary tract infections lacked an elevation of serum IL-6 levels. Furthermore, serum C-reactive protein and body temperature were also not related to urinary IL-6 levels [207]. The prognostic value of urinary IL-6 levels was substantiated in a recent meta-analysis that identified urinary IL-6 concentrations as appropriate to detect febrile urinary tract infections in children and to distinguish them from other febrile conditions [208] (Figure 5). In contrast to these findings, an earlier study could not detect significant differences in urinary IL-6 levels when comparing nonbacteremic and bacteremic febrile patients. The latter study concluded that IL-6 is produced in the urinary tract regardless of bacteremia to trigger the host immune response [209]. Non-invasive testing as an alternative to serum is especially important in children [210]. Urinary IL-6 levels may have the potential to monitor therapeutic success in children with urinary tract infections.

#### 2.6.3. Urinary IL-6 in SLE

Urinary IL-6 levels of 16 SLE patients were higher compared to controls (n = 34) and correlated with disease activity, which was assessed by the Systemic Lupus Activity Measure [211] (Figure 5). Stratification of 63 patients into groups displaying low (<7.3 pg/mL) and high (>7.3 pg/mL) urinary IL-6 levels revealed that the latter cohort had shorter disease duration and higher proteinuria. Systemic Lupus Activity Measure and eGFR did not differ between these two groups [212]. This suggested that impaired kidney function contributes to higher urinary IL-6 levels in SLE. In support of this, patients with lupus nephritis had indeed increased urinary IL-6 concentrations compared to patients with normal kidney function [213].

Urinary IL-6 levels were associated with renal disease activity assessed by the renal SLE disease activity index [213]. Along these lines, urinary samples of patients with lupus nephritis contained significant IL-6 activity, and IL-6 mRNA was expressed in the glomeruli of patients [214]. Serum IL-6 levels did not change in SLE [211] and lupus nephritis [213]. Hence, in lupus nephritis, locally produced IL-6 may emerge as a biomarker for disturbed kidney function.

#### 2.6.4. Urinary IL-6 in IgA Nephropathy

IgA nephropathy is a common form of primary glomerulonephritis and a leading cause of chronic kidney disease. These patients exhibit increased urinary IL-6 levels in comparison to healthy controls [214,215] (Figure 5).

IL-6 mRNA was found to be expressed in the glomeruli of these patients, suggesting that renal synthesis of IL-6 may contribute to higher urinary IL-6 levels [214]. Mesangial cell proliferation is the underlying pathophysiological feature of renal diseases such as IgA nephropathy, SLE and diabetic nephropathies [216], and markedly, the amount of urinary IL-6 correlated with the degree of cell proliferation [217], which stimulated the growth of rodent mesangial cells [214]. Three of the twenty-one patients developed progressive IgA nephropathy disease, which correlated with very high urinary IL-6 levels, suggesting that determination of urinary IL-6 concentrations may serve as a prognostic factor for the development of IgA nephropathy [215] (Figure 5). In line with this, the follow-up of patients with high and low urinary IL-6 levels for 10 months showed histological progression only in the first group [218]. Furthermore, patients characterized by a decline in urinary IL-6 levels during the study period displayed histologic improvement of the disease [218]. Notably, corticosteroid therapy did improve proteinuria, but did not lower urinary IL-6 levels [215]. On the other hand, a patient with sarcoidosis and high urinary IL-6 levels showed a decline in IL-6 concentration in the urine after prednisolone therapy, which also improved renal function [219]. Hence, urinary IL-6 levels could be a valuable prognostic marker in kidney disease progression.

#### 2.6.5. Urinary IL-6 in Diabetic Nephropathy

Kidney damage is commonly observed in diabetics, yet serum and urinary IL-6 levels did not differ between controls and type 2 diabetic patients with normal kidney function. Yet, serum as well as urinary IL-6 levels increased in patients with type 2 diabetes and proteinuria (Figure 5). In this study, urinary IL-6 levels were, however, not related to serum IL-6 concentrations [220]. Similar to the results described above for IgA nephropathy and lupus nephritis (Section 2.6.3 and Section 2.6.4), these findings further substantiate the hypothesis that kidney dysfunction is related to higher urinary IL-6 levels.

#### 2.6.6. Urinary IL-6 in Organ Rejection

The best treatment for kidney failure is organ transplantation. However, the incidence of organ rejection is high, and non-invasive biomarkers for early diagnosis of rejection are required [221]. Given the marker potential of IL-6 levels in the urine as a read-out for kidney function (see above Section 2.6.3, Section 2.6.4 and Section 2.6.5), IL-6 concentrations were monitored over the first 30 postoperative days in 40 patients receiving renal transplants. Indeed, IL-6 levels were higher in the 26 patients who had episodes of acute rejection than in those without this complication (Figure 5). Doubling of urinary IL-6 levels within 24 h occurred in patients with acute organ rejection within the following two days [222]. These studies provide strong support for urinary IL-6 levels to emerge as an early biomarker for kidney rejection.

#### 2.6.7. Fecal IL-6 in Gastroenteritis

Rotavirus and norovirus are common causes of viral gastroenteritis in children. Fecal IL-6 levels correlated with the number of daily bowel movements [223] and were also elevated in children with acute gastroenteritis [224] (Figure 5). Based on these promising findings, additional studies will have to further validate the diagnostic potential of fecal IL-6 levels in gastrointestinal diseases.

#### 2.6.8. Fecal IL-6 in Liver Cirrhosis

Patients with liver cirrhosis are characterized by increased serum IL-6 levels due to a higher synthesis of IL-6 in immune cells as well as an impaired hepatic removal of this cytokine [2,15]. Decompensated liver cirrhosis is often complicated by ascites, and circulating IL-6 levels are further increased in these patients [2]. Besides the build-up of fluid (ascites), acute decompensation of liver cirrhosis is also accompanied by variceal hemorrhage, hepatic encephalopathy and/or bacterial infection. Fecal as well as plasma IL-6 levels were higher in patients with acutely decompensated liver cirrhosis but were similar between healthy controls and patients with stable liver cirrhosis [225] (Figure 5). Hence, fecal IL-6 levels may be suitable as an early biomarker of decompensation in patients with liver cirrhosis.

## 3. Conclusions

In this review, we have summarized current knowledge of and challenges in detecting adipokines in urine and feces. In fact, human urine and feces contain measurable levels of adiponectin, lipocalin-2, leptin, galectin-3, chemerin and IL-6, with leptin and chemerin levels being relatively low in stool. In regard to their potential to identify disease and its progression, patients with kidney diseases have higher levels of urinary adiponectin, lipocalin-2, leptin, galectin-3 and IL-6 (Table 1) [67,130,156,174,212]. Hence, these adipokines may emerge as biomarkers for the diagnosis and prognosis of kidney diseases. It is important to note that bacterial infections may affect the urinary levels of adipokines [88,104,207], and this may limit the immediate clinical value of these biomarkers. The upregulation of adipokine synthesis in injured kidney cells or higher adipokine production in leukocytes and/or elevated circulating adipokine levels together with increased renal excretion rates seem to contribute to their urinary levels, and evaluation of the pathways mediating elevated urinary concentrations of these adipokines are of pathophysiological relevance.

Biomarkers for the diagnosis and monitoring of disease activity in patients with inflammatory bowel diseases are urgently needed. Relevant in this context could be chemerin and lipocalin-2 levels, which are induced in patients with active disease but do not greatly vary between patients with different disease etiologies. Notably, low fecal galectin-3 levels were found to be related to a worse disease pathology [22,139,180,196] (Table 1). Non-invasive biomarkers to monitor the therapeutic success of cancer therapy are of clinical value, and urinary galectin-3 is an interesting protein in this regard [177] (Table 1).

Despite these promising findings, the clinical suitability of the urinary and fecal adipokines described in this review article has to be evaluated in future studies. Alternatively, a further option for the identification of disease-specific variations of adipokines could be a comparative analysis of macromolecular adipokine complexes and adipokine isoforms. Limitations for the use of these biomarkers are manifold. The stability of these proteins in urine and feces as well as storage and sample preparation conditions have mostly not been evaluated. Time of sampling and normalization of the values by, e.g., urinary creatinine levels may be important and have to be standardized. In summary, the development and standardization of assays to determine urinary and fecal adipokine levels, and combining those readings with other biomarkers, may provide potential benefits for disease prediction, diagnosis and therapy.

Further complexity in the interpretation of urinary adipokine levels may arise from the fact that their serum concentrations can differ between sexes and are often related to BMI and metabolic diseases [2,21,48,50,151,165,226]. Commonly used drugs such as atorvastatin may promote intestinal barrier dysfunction and may also affect fecal and urinary protein levels [227]. However, how these parameters may influence urinary or fecal adipokine levels has not yet been thoroughly analyzed, and further research is required to clarify these issues. Adipokines have been the subject of much scientific work, but more research is needed for implementation in clinical routines.

## Figures and Tables

**Figure 1 biomedicines-11-01186-f001:**
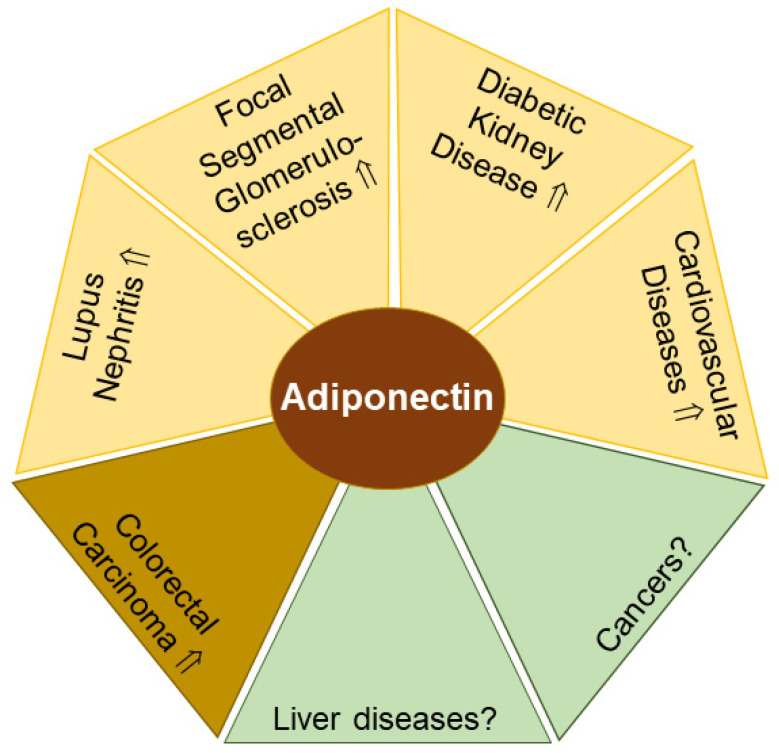
Adiponectin levels in urine (yellow) and feces (light brown) have potential to serve as biomarkers for the indicated diseases. In liver diseases and cancer (green) with altered circulating adiponectin levels, the diagnostic and prognostic value of urinary and/or fecal adiponectin remains to be clarified. Elevated (⇑) adiponectin levels and a lack of data on adiponectin amounts (?) in urine or feces in patients compared to the respective controls are indicated.

**Figure 2 biomedicines-11-01186-f002:**
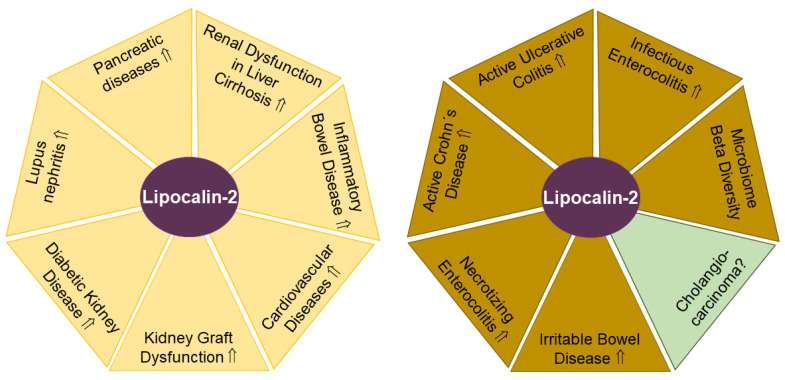
Lipocalin-2 levels in urine (yellow areas) and feces (light brown areas) have potential to serve as biomarkers for the indicated diseases. In cholangiocarcinoma (green) with altered biliary lipocalin-2 levels, the diagnostic and prognostic value of fecal lipocalin-2 remains to be clarified. Elevated (⇑) lipocalin-2 levels and a lack of data on lipocalin-2 amounts (?) in urine or feces in patients compared to the respective controls are indicated.

**Figure 3 biomedicines-11-01186-f003:**
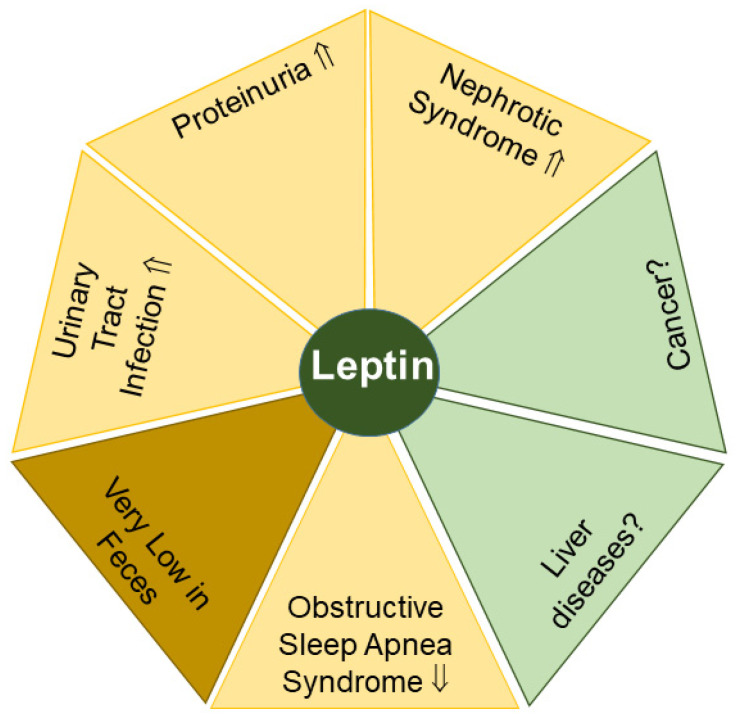
Leptin levels in urine (yellow) may serve as possible biomarkers for the indicated diseases. Significant amounts of leptin in feces (light brown) are difficult to detect. Elevated (⇑) and reduced (⇓) leptin levels and a lack of data (?) on leptin levels in urine or feces in patients compared to the respective controls are indicated.

**Figure 4 biomedicines-11-01186-f004:**
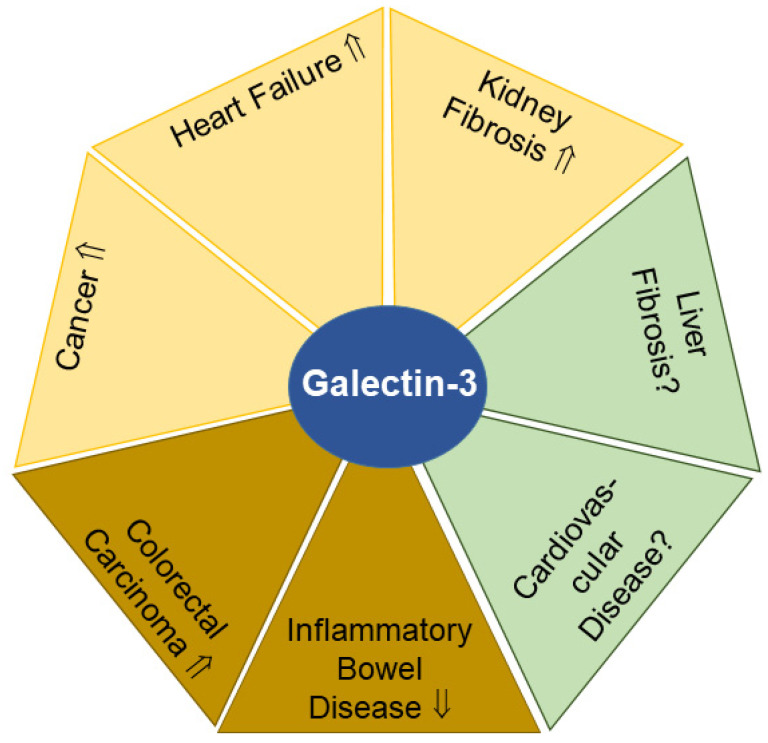
Galectin-3 levels in urine (yellow) and feces (light brown) have potential to serve as biomarkers for the indicated diseases. In liver fibrosis and cardiovascular diseases (green), the diagnostic and prognostic value of urinary and/or fecal galectin-3 remains to be clarified. Elevated (⇑) or reduced (⇓) galectin-3 levels and a lack of data on galectin-3 amounts (?) in urine or feces in patients compared to the respective controls are indicated.

**Figure 5 biomedicines-11-01186-f005:**
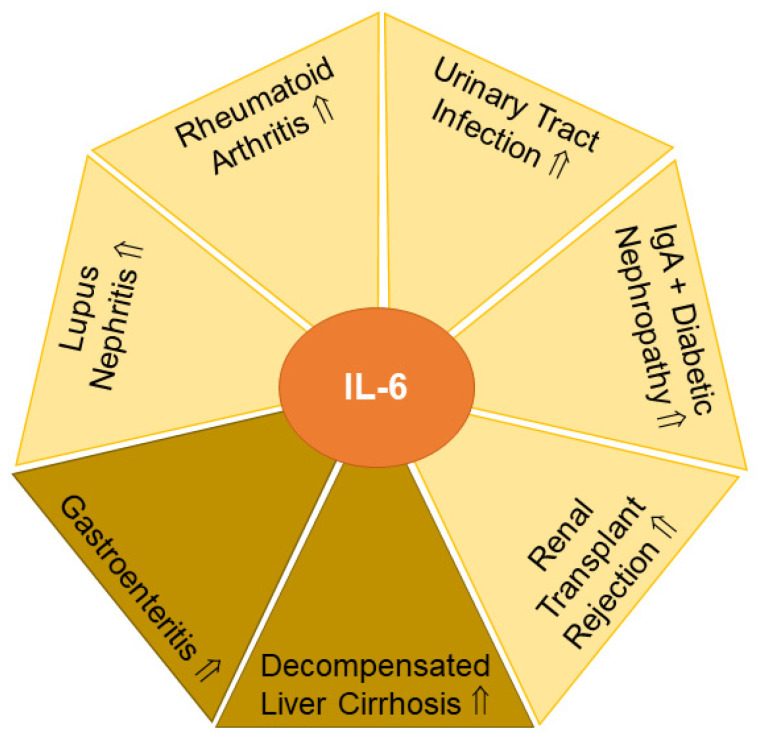
IL-6 levels in urine (yellow) and feces (light brown) have potential to serve as biomarkers for the indicated diseases. Elevated (⇑) IL-6 levels in urine or feces in patients compared to the respective controls are indicated.

**Table 1 biomedicines-11-01186-t001:** Urine and/or feces levels of adiponectin, chemerin, galectin-3, IL-6, leptin and lipocalin-2 have potential to serve as biomarkers for the diseases listed in the left column. Elevated (⇑) and reduced (⇓) levels in urine or feces in patients compared to the respective controls are listed in the right column. Diseases as well as adipokines are listed in alphabetical order.

Disease	Urinary/Fecal Biomarkers
Cancer	Urinary Galectin-3 ⇑
Cardiovascular Disease	Urinary Adiponectin ⇑ Urinary Lipocalin-2 ⇑
Colorectal Carcinoma	Fecal Adiponectin ⇑ Fecal Galectin-3 ⇑
Decompensated Liver Cirrhosis	Fecal IL-6 ⇑
Diabetic Kidney Disease	Urinary Adiponectin ⇑ Urinary IL-6 ⇑ Urinary Lipocalin-2 ⇑
Focal Segmental Glomerulosclerosis	Urinary Adiponectin ⇑
Gastroenteritis	Fecal IL-6 ⇑
Heart Failure	Urinary Galectin-3 ⇑
IgA Nephropathy	Urinary IL-6 ⇑
Infectious Enterocolitis	Fecal Lipocalin-2 ⇑
Inflammatory Bowel Disease	Urinary Chemerin ⇑ Fecal Galectin-3 ⇓ Fecal Lipocalin-2 ⇑ Urinary Lipocalin-2 ⇑
Irritable Bowel Disease	Fecal Lipocalin-2 ⇑
Kidney Fibrosis	Urinary Galectin-3 ⇑
Kidney Graft Dysfunction	Urinary IL-6 ⇑ Urinary Lipocalin-2 ⇑
Liver Cirrhosis-Related Kidney Dysfunction	Urinary Lipocalin-2 ⇑
Lupus Nephritis	Urinary Adiponectin ⇑ Urinary IL-6 ⇑ Urinary Lipocalin-2 ⇑
Necrotizing Enterocolitis	Fecal Lipocalin-2 ⇑
Nephrotic Syndrome	Urinary Leptin ⇑
Obstructive Sleep Apnea Syndrome	Urinary Leptin ⇓
Pancreatic Diseases	Urinary Lipocalin-2 ⇑
Proteinuria	Urinary Leptin ⇑
Rheumatoid Arthritis	Urinary IL-6 ⇑
Urinary Tract Infection	Urinary IL-6 ⇑ Urinary Leptin ⇑

## Data Availability

Not applicable.

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
