# Peer review of "Fecal and Urinary Adipokines as Disease Biomarkers"

_biomedicines, 2023, doi:10.3390/biomedicines11041186_

Round 1

Reviewer 1 Report

The interesting review about selected adipokines in urine and feces which may emerge as biomarkers of different disease. The comparison with serum levels gives the added value to this article and potential of these biomarkers to serve as diagnostic and prognostic tool.

Author Response

Thank you so much for your kind comments. According to the comments of Reviewer 1 and 2 we now included a table summarizing the biomarkers described in the article at the end of the manuscript.  

Reviewer 2 Report

The identification and utilization of adipokine levels as urinary and fecal biomarkers can offer a significant advantage for disease diagnosis and predicting treatment outcomes, as analyzing urine and feces from patients is cost-efficient and non-invasive. This review article emphasizes the abundance of selected adipokines in urine and feces, highlighting their potential to serve as diagnostic and prognostic biomarkers. The article is well-written, and there are some problems needed to be adressed.

1.The author selected adiponectin, lipocalin-2, leptin, galectin-3, chemerin, and interleukin-6 as the adipokines of interest. However, the introduction should describe why only these few were chosen.

2. Please provide tables that succinctly summarize the clinical results of these biomarkers, making it easier for auditors to review.

3. Do these biomarkers have any limitations or drawbacks that can be further discussed in the article?

Author Response

We are very grateful to the reviewer for very kind comments on our manuscript and the points raised which helped us to improve this review article.

1.The author selected adiponectin, lipocalin-2, leptin, galectin-3, chemerin, and interleukin-6 as the adipokines of interest. However, the introduction should describe why only these few were chosen.

Thank you for this important advice. Indeed we searched PubMed with various tags but mostly were not successful. We added a short explanation at the end of the Introduction.

  1. Please provide tables that succinctly summarize the clinical results of these biomarkers, making it easier for auditors to review.

We now added one table at the end of the article summarizing all of the biomarkers in relation to the disease. Adding only one table has the advantage to more easily identify different biomarkers for specific diseases. We hope you are satisfied with this change.

  1. Do these biomarkers have any limitations or drawbacks that can be further discussed in the article?

We already discussed some limitations in the article. Please see e.g. page 8: “Critically ill patients with acute kidney injury also displayed high urinary lipocalin-2 levels and this was related to disease progression and mortality. However, with urinary lipocalin-2 levels being elevated in patients with sepsis, urinary infections or cardiopulmonary bypass surgeries, the prognostic utility of urinary lipocalin-2 concentrations for acute kidney injury appears limited, which is especially relevant for patients in intensive care.

Or in Conclusion

““Further complexity on the interpretation of urinary adipokine levels may arise from the fact that their serum concentrations can differ between sexes, and are often related to BMI and metabolic diseases.”

To address this further we added a short paragraph in Conclusions.   

Reviewer 3 Report

The manuscript by Tews et al. summarize the current understanding on the major adipokines detected in the urine and faeces, focusing on the clinical relevance of these biomarkers in the urine and faeces in relation to diseases. Overall the review is comprehensive, well organized and provides adequate information in the field. The topic is new which disguises itself with most of the other ones which mainly focus on the serum level of the adipokines. A few issues can be rectified to further improve the manuscript.

1. A table shall be made to summarize and list all the adipokines included in the manuscript.

2. Though the clinical studies on urinary and faecal adipokines are extensively summarized, whether or not there currently exist animal studies to further interrogate the pathological function of these adipokines? 

Author Response

We are very grateful to the reviewer for very kind comments on our manuscript and the points raised which helped us to improve this review article.

  1. A table shall be made to summarize and list all the adipokines included in the manuscript.

We added a table at the end of the review.

  1. Though the clinical studies on urinary and faecal adipokines are extensively summarized, whether or not there currently exist animal studies to further interrogate the pathological function of these adipokines? 

Thank you for your kind comments.

Answer to your question related to animal studies is that the pathophysiological function of these adipokines has been extensively studied. A PubMed search for “adiponectin mice” found 5,026 articles, ”leptin mice” 9,869  articles,  “IL-6 mice” 51,361 articles, “galectin-3 mice” 1.173 article and “lipocalin-2 mice” 962 articles. We think it is beyond the scope of this article to summarize even these animal studies related to the human disease described. We hope you agree with that.